# Fetal Exposure to Endocrine Disrupting-Bisphenol A (BPA) Alters Testicular Fatty Acid Metabolism in the Adult Offspring: Relevance to Sperm Maturation and Quality

**DOI:** 10.3390/ijms24043769

**Published:** 2023-02-13

**Authors:** Saikanth Varma, Archana Molangiri, Suryam Reddy Kona, Ahamed Ibrahim, Asim K. Duttaroy, Sanjay Basak

**Affiliations:** 1Molecular Biology Division, ICMR-National Institute of Nutrition, Indian Council of Medical Research, Hyderabad 500007, India; 2Lipid Chemistry Division, ICMR-National Institute of Nutrition, Indian Council of Medical Research, Hyderabad 500007, India; 3Department of Nutrition, Institute of Basic Medical Sciences, Faculty of Medicine, University of Oslo, 0316 Oslo, Norway

**Keywords:** testis, PUFA, BPA, male fertility, desaturase, PPAR

## Abstract

Daily exposure to bisphenols can affect reproductive functions due to their pseudo-estrogenic and/or anti-androgenic effects. Testicular lipids contain high levels of polyunsaturated fatty acids necessary for sperm maturity, motility, and spermatogenesis. Whether prenatal exposure to bisphenols alters testicular fatty acid metabolism in adult offspring is unknown. Pregnant Wistar rats were gavaged from gestational day 4 to 21 with BPA and BPS (0.0, 0.4, 4.0, 40.0 μg/kg bw/day). Despite increased body and testis weight, the total testicular cholesterol, triglyceride, and plasma fatty acids were unaffected in the offspring. Lipogenesis was upregulated by increased SCD-1, SCD-2, and expression of lipid storage (ADRP) and trafficking protein (FABP4). The arachidonic acid, 20:4 n-6 (ARA) and docosapentaenoic acid, 22:5 n-6 (DPA) levels were decreased in the BPA-exposed testis, while BPS exposure had no effects. The expression of PPARα, PPARγ proteins, and CATSPER2 mRNA were decreased, which are important for energy dissipation and the motility of the sperm in the testis. The endogenous conversion of linoleic acid,18:2 n-6 (LA), to ARA was impaired by a reduced ARA/LA ratio and decreased FADS1 expression in BPA-exposed testis. Collectively, fetal BPA exposure affected endogenous long-chain fatty acid metabolism and steroidogenesis in the adult testis, which might dysregulate sperm maturation and quality.

## 1. Introduction

Global decline in fertility rate and increasing male infertility could originate from the fetus. Testicular dysgenesis syndrome is suspected due to adverse environmental exposure to fetal life gonadal programming [1]. Exposure to endocrine-disrupting chemicals (EDC), such as bisphenols, is inevitable in our everyday life, even at the microscopic level, through various modes, including dermal exposure, leaching from food-containing surfaces, microplastic contaminants from marine sourced foods, and the recycling of non-degradable residual pollutants. Although bisphenol A (BPA) is detected in maternal and fetal tissues [2], the fetus is not equipped with BPA-antagonizing machinery, making it vulnerable to exposure during the developmental window, which could have long-lasting programming on the functions of organs. BPA binds to estrogen-related receptor gamma with high affinity and suppresses androgen receptors in the testis, increasing the risk for male infertility [3]. Emerging data suggest a closer alignment between urinary BPA levels and reproductive malfunctions in males [4]. In utero BPA exposure alters the functioning of the male reproductive system in the offspring [5], affecting testicular physiology [6], disrupting its bioenergetics capacity [7], and metabolism [8]. Testicular metabolism is tightly regulated by each cell type performed within its microenvironment. A detailed understanding of the potential risk associated with fetal exposure to BPA on testicular physiology could help identify underlying causes of declining male fertility.

Lipids are essential in sperm membrane formation since their compositional changes during testis maturity determine sperm motility and affect spermatogenesis [9]. Spermatogenesis could also be influenced by an interplay of hormone-disrupting substances, such as bisphenols, and membrane lipid biosynthesis of sperm since exposure to BPA mimics high-fat diet (HFD)-induced metabolic changes in the offspring [10]. Exposed to a pollutant mixture containing BPA, they exhibited dysregulated lipid homeostasis with increased hepatic triglycerides and a few similarities with HFD-induced changes, particularly regarding lipid metabolism [11], indicating the modulation of the endocrine–metabolic axis by BPA. Since the structural requirement of maturing testis demands increased lipid biosynthesis, their turnover increases during development. Spermatids account for more than 60% of the total testis volume in rats, with fatty acid concentrations similar in both testis and spermatids. Fatty acid composition determines the performance not only of sperm maturity but also of acrosomal reaction and sperm–oocyte fusion [12]. The fatty acid composition in the testis is associated with male fertility [13,14], affecting sperm maturity due to the non-availability of the substrates to the testis [12]. Understanding the impacts on testis fatty acid metabolism due to exposure to bisphenols could help identify a potential cause of male infertility. Even though fetal BPA exposure to fatty acid composition in the liver and adipose [8] is reported, the effect of in utero exposure to BPA and its substitute BPS on testicular lipid metabolism in adult offspring is not known.

In addition to the liver and other fatty acid-synthesizing tissues, the testis is the site for the active biosynthesis of polyunsaturated fatty acids (PUFAs). The activities of desaturases and elongases regulate it. The testicular fatty acids profile affects their cells and influences sperm maturity, capacitation, acrosome reactions, and sperm–oocyte fusion, contributing to male fertility [12]. The metabolism of PUFAs harnesses energy for spermatogenesis and epididymal sperm maturation. Fatty acids, especially long-chain (LC) PUFAs, play a vital role in spermatogenesis. Moreover, disrupted fatty acid composition produces infertile sperm, affecting sperm maturation and motility. Increased MUFAs and decreased PUFAs in spermatozoa are correlated with male infertility [13]. Very long-chain PUFAs in the testis positively correlate with sperm quality and are involved with improved sperm quality [14].

PUFAs influence male reproductive functions in several ways. They are the precursors of prostaglandin (PG) synthesis and modulate the expression of key enzymes involved in steroid and fat metabolism of the testis. Spermatozoa need plasma membrane PUFA enrichment to support membrane fluidity during fertilization. However, it also makes spermatozoa susceptible to attack by reactive oxygen species. BPA exposure promotes oxidative stress in the testis, thereby establishing a potential association with reduced fertility. Eicosanoid biosynthesis involves testicular development by supporting PG synthesis, which is needed for germ cell development and steroidogenesis. Arachidonic acid, 20:4 n-6 (ARA), is a precursor for the PG-2 series during testicular steroidogenesis in spermatogenesis [15]. Although the involvement of PGs is well-established in female reproduction, their role in male fertility was later evidenced by dysregulated cyclooxygenase-2 (COX2) expression in infertile subjects [16].

Fatty acid composition is influenced by diet and endogenous desaturation activities in the testis. The LCPUFAs, such as docosapentaenoic acid (DPA, 22:5, n-6) and docosahexaenoic acid (DHA, 22:6, n-3) are essential for sperm quality. The local accumulation of these LCPUFAs in spermatids supports spermatogenesis [17]. The fatty acid composition of testicular spermatozoa determines sperm characteristics [18]. LCPUFAs are synthesized locally in Sertoli cells from their precursor, uptaking, and transported to the testis germ cell. Sertoli cells contain much higher fatty acid elongation and desaturation activities than germ cells [19]. Unlike other PUFA-rich tissue, fatty acids are continuously utilized in the testis as spermatozoa are transported to the epididymis. All the major desaturases, such as FADS1, FADS2, SCD1, and SCD2, are localized in the Sertoli cells of the rat testis [20]. The activity of desaturases is involved in the endogenous biosynthesis of monounsaturated fatty acids (MUFAs), PUFAs, and LCPUFAs from their precursors by SCDs, FADSs, and ELOVLs enzymes. Despite evidence that suggests SCDs [8,21] and FADSs are altered in male fertility and spermatogenesis [12], prenatal bisphenol exposure to testicular PUFA metabolism and its relation with FADS activities in the developmentally exposed offspring is elusive.

Spermiogenesis is an energy-intensive process transforming round spermatids into elongated spermatozoa for fertile sperm production. Site-specific metabolic fuel utilization supports reproductive energy requirements during spermatogenesis. Peroxisome proliferator-activated receptor gamma (PPARγ) is a master regulatory switch for the energy metabolism of fats, predominantly expressed in the germ cells and Sertoli cells of the testis, involved in regulating the expression of lipid metabolic genes by forming PPARγ-RXRα heterodimer in Sertoli cells [22], and its signalling contributes a role in energy metabolism in the testis. Studies reported that lipid metabolic genes in the testis regulates male fertility [22] and plays a role in male reproduction by binding to DNA promoters of the genes for fatty acid metabolism [23]. Despite these data, the mechanism by which PPARs regulate energy metabolism in the testis during endocrine disruption is unclear.

Fatty-acid-binding proteins (FABPs) involve intracellular fatty acid trafficking and metabolism to support the synthesis and remodelling of membrane lipids in the testis [24]. FABP9 is expressed in the interstitial tissue of the testis, a major component of the rat sperm perinuclear theca. It plays a protective role against the oxidation of sperm fatty acids and sperm function. Since expressions of FABPs are involved in the testicular lipid metabolism and functionally involved in the final stages of germ cell differentiation in the testis, it needs to be investigated.

Excess adiposity affects the fatty acid composition of spermatozoa via regulating testis fatty acid metabolism [18]. Fetal BPA exposure results in elevated triglycerides, adiposity, and adipogenic gene expression in developmentally exposed rodent offspring [8,25]. However, studies investigating the fetal programming of testicular fatty acid metabolism and its functions following developmental exposure to BPA are limited.

Our recent data demonstrated that prenatal exposure to bisphenols (BPA and BPS) significantly increased body and testis weight and altered testosterone levels, androgen, estrogen receptor expression, hyperplasia, inflammation, and oxidative stress in the offsprings’ testis [5]. We hypothesize that lipogenesis in the testis may augment hyperplasia and increase vacuolization. Assessing the effects of in utero BPA exposure on the fatty acids profile of the offspring testis could help identify the risk of exposing lifestyle factors to reproductive functions. Therefore, the effects of in utero bisphenol exposure on testicular fatty acid metabolism of the adult offspring were investigated. The study conducted in rodents may represent a model for understanding lipid metabolism in the human testis, particularly when fetal gonadal development is exposed to EDC. The study reports the impact of bisphenol exposure on the total PUFA composition, the metabolic pathway of LCPUFAs, localization, and expression of active desaturase, PPARs, and fatty-acid-binding proteins in adult offspring testis exposed to bisphenols prenatally.

## 2. Results

### 2.1. In Utero, BPA Exposure Affects Long-Chain Fatty Acid Composition in the Testis Independent of Plasma

Whether lipogenesis and/or changes in fatty acid composition in the testes are the probable underlying reasons for the testicular hyperplasia observed previously was examined by measuring their levels in plasma and testis. The plasma total fatty acid composition of the adult offspring did not report any significant changes in the saturated fatty acids (SFA), monounsaturated fatty acid (MUFA), PUFA, and LCPUFA levels across the experimental groups involved with various concentrations (0.0, 0.4, 4.0, and 40.0 µg/kg bw/day) of bisphenols (BPA and BPS; Table 1) exposed during gestation. In contrast, the total fatty acid composition was significantly altered in the offspring’s testes exposed to BPA (4.0 µg/kg bw/day) in utero (Table 2). The concentration of SFAs (palmitic acid and stearic acid) decreased (*p* < 0.05), while MUFAs (palmitoleic acid and oleic acid) and PUFA (linoleic acid) increased (*p* < 0.05) in BPA-exposed (4.0 µg/kg) testis. While in the control testis, LA was converted efficiently to produce ARA and DPA and increased their levels by ~2.9 and ~3.2 folds, but their levels were decreased by ~5.8 and ~9.6 folds, respectively, in BPA-exposed testis (Table 2).

Stearoyl-coenzyme A desaturases (SCDs) or delta-9-desaturase are involved in the biosynthesis of MUFA from SFAs by introducing a double bond. Therefore, the SCD indexes (SCD-16 and SCD-18) were estimated by the product/precursor ratios of the fatty acids. The increased SCD-16 and SCD-18 in the BPA-exposed testis (Table 2) reflect a higher SCD-1 activity. Increased SCD indexes indicate that fetal BPA exposure might have programmed testicular lipid homeostasis, increasing lipogenesis in adult offspring testis. Overall, fetal exposure to BPA affects the testicular lipid metabolism of adult offspring by modulating SCD-1 activities.

### 2.2. LCPUFAs Were Decreased in BPA-Exposed Testis, While BPS Exposure Had No Effects

LCPUFAs such as arachidonic acid, 20:4 n-6 (ARA), adrenic acid (C22:4 n-6), and docosapentaenoic acid, C22:5 n-6 (DPA) were significantly decreased by ~3.4, ~2.0, and ~6.3 folds (*p* < 0.05). In contrast, the levels of oleic acid, C18:1 (OA), and linoleic acid, C18:2 (LA), were significantly increased (*p* < 0.05) by ~2.4 and ~4.9 folds, respectively, only in BPA-exposed offspring (Table 2). However, the level of docosahexaenoic acid, C22:6 n-3 (DHA), was unaffected in testis among these groups. Whether the increased lipid levels in these offspring were due to an increase in total cholesterol and triglycerides, their levels were measured in the plasma and testes of the adult offspring. In utero, BPA exposure did not alter the triglycerides and total cholesterol in the plasma and testes of 90 d adult offspring (Table 3). Collectively, these data indicate that maternal exposure to BPA, not BPS during gestation, increases testis fatty acid composition and lipogenesis in the rats at 90 days of age.

### 2.3. Expression of Fatty Acid Storage, Lipogenesis and Sperm Motility Mediator Was Affected in the Testis Due to BPA Exposure

To understand the mechanism of disrupted lipid composition in offspring testes, the expression of the proteins associated with fatty acid storage, trafficking, and metabolism, such as ADRP, FABP4, FABP5, FABP7, and FABP3, were measured in the testis homogenate obtained from 90 d male rat offspring. The expressions of ADRP and FABP4 were significantly increased by ∼5.6 and ∼5.3 folds (*p* < 0.05) in BPA-exposed testis (Figure 1A,B), while the expression of FABP3 was unchanged, and FABP5 and FABP7 were not detected in the testis. Taken together, these data suggest that BPA exposure during pregnancy profoundly affected the offspring’s testicular expression of lipid storage and trafficking proteins in their adult life.

To examine the mechanisms associated with lipogenesis, the mRNA expression of the genes that regulate lipogenesis and steroidogenesis was investigated in the adult testis. In utero, BPA exposure significantly upregulated hormone-sensitive lipase (LIPE) expression by ~1.35 fold (*p* < 0.05). In contrast, the expression of lipoprotein lipase (LPL) was decreased (*p* > 0.05) in the offspring’s testis (Figure 2). The expressions of testis-specific fatty-acid-binding protein 9 (FABP9) and cyclooxygenase 2 (COX2) mRNAs were significantly downregulated (*p* < 0.05) in the rat testis exposed to BPA (Figure 2). The expression of genes involved in lipogenesis, such as fatty acid synthases (FASN) and mitochondrial fatty acid transporter (SLC25A20), remained unchanged in the testis when compared to the control. The expression of growth and signalling molecules, such as insulin growth factor-1 (IGF-1), leptin (LEP), and adiponectin (ADIPOQ), decreased marginally (*p* > 0.05) in BPA-exposed rat testis.

Testicular lipid turnover is critically involved in successful spermatogenesis. The expressions of nuclear receptor proteins, such as PPARα and PPARγ, are associated with fatty acid metabolism and energy homeostasis in spermatogenesis; therefore, they were measured in the testis homogenate prepared from 90 d male rat offspring. The expression of PPARγ and PPARα significantly decreased by ∼1.6 and ∼1.9 folds (*p* < 0.05) in BPA-exposed testis (Figure 3A,B) as compared to the control rats. Thus, BPA exposure in utero significantly affected the expression of PPAR-signalling proteins in the testis, which may regulate sperm energy dissipation. In addition, BPA exposure to dams during pregnancy significantly downregulated CATSPER2 mRNA expression (*p* < 0.05), but the expression of CATSPER1 had no significant effects compared to the control (Figure 4).

### 2.4. Stearoyl-CoA Desaturase Activities, and Fatty Acid Desaturase & Elongase Expression Were Altered in Adult Testis Due to In Utero BPA Exposure

Since LCPUFA biosynthesis was severely affected in the testes, the expression of fatty acid desaturase 1 (FADS1 or delta-5 desaturase), fatty acid desaturase 2 (FADS2 or delta-6 desaturase), delta 4 desaturase sphingolipid 1 (DEGS1), the elongation of very long-chain fatty acids-like 2 (ELOVL2), the elongation of very long-chain fatty acids-like 5 (ELOVL5), stearoyl-coenzyme A desaturase 2 (SCD2), and stearoyl-coenzyme A desaturase 1 (SCD1) were screened quantitatively. The mRNA expression of fatty acid desaturase 1 (FADS1) was significantly downregulated (*p* < 0.05) in the testis of the BPA-exposed (4.0 µg/kg) rats compared to the control (Figure 5). The expression of other desaturases, such as DEGS1, SCD1, and SCD2, also decreased (*p* > 0.05) in the testes of these rats. However, the mRNA levels of FADS2, ELOVL2, and ELOVL5 remained unaltered due to BPA exposure. 

Further, the expression of the enzymes involved in PUFA biosynthesis, such as FADS1 and ELOVL2 proteins, were measured in the testis homogenate prepared from 90 d male rat offspring. In the testes, the expression of FADS1 significantly decreased by ~4.3 folds (*p* < 0.05, Figure 6A), while ELOVL2 expression increased by ~1.4 folds (*p* < 0.05, Figure 6B) in BPA-exposed rats. BPA exposure in utero significantly lowered the FADS1 expression with compensatory expression of ELOVL2, indicating a disruption in the desaturation and elongation activities associated with LCPUFA biosynthesis in these adult offspring testes.

### 2.5. FADS1 Proteins Are Dispersed and Scattered in the Seminiferous Tubules of the Testis Due to BPA Exposure

The localization and expression of FADS1 were examined by using immunofluorescence on the offspring testes, which were gestationally exposed to BPA (4.0 µg/kg). Green dots indicate the expression and localization of FADS1 protein in the testicular tissue obtained from the control and BPA-exposed rats (Figure 7A). The normalized corrected total cell fluorescence (CTCF) of FADS1 was significantly decreased in the BPA-exposed (4.0 µg/kg) testis compared to the control (control vs. BPA: 35.42 ± 2.49 vs. 11.69 ± 0.67, *p* < 0.001, Figure 7B). The FADS1 signal (green) was observed predominantly around the lumen of the seminiferous tubule, which is rich with elongated spermatids and Sertoli cells. Scattered and decreased signals of FADS1 demonstrates disrupted activities and the reduced expression of the fatty acid desaturase 1 enzyme in the adult rat testes that were gestationally exposed to BPA.

## 3. Discussion

For the first time, this study reports that the TDI level (4.0 µg/kg bw/day) of prenatal BPA exposure affected lipogenesis, stearoyl-CoA desaturase (delta-9-desaturase) index, the levels of MUFAs and n-6 LCPUFAs, and fatty acid desaturase 1 (FADS1 or delta-5 desaturase) expression in adult offspring testis, which in turn may be responsible for the dysregulation of sperm maturity. The PUFA composition of the whole testis showed a reduced proportion of n-6 LCPUFAs in the BPA-exposed testis than the controls, where ARA (20:4 n-6) and DPA (22:5 n-6) levels were dramatically decreased. The lower content of these n-6 LCPUFAs can reduce membrane fluidity [26] in the BPA-exposed offspring testis and, thus, consequently, affect the maturity of the sperm. The localization and reduced expression of FADS1 in the adult testes indicates that gestationally, BPA exposure affected fatty acid desaturase machinery in these rats.

Decreased levels of n-6 LCPUFA may affect spermatogenesis by reducing prostaglandin (PG) production. The cyclooxygenase (COX) enzyme regulates gonadal development via PG in the testis [15]. ARA, liberated from phospholipids, is rapidly converted to PGs by COX enzymes and thus stimulates testicular steroidogenesis during spermatogenesis. The decreased availability of ARA in BPA-exposed offspring testis (Table 2) can affect fertilization due to the reduced production of PGE2, which plays a major role in sperm survival and maturity. The suppressed testicular PGE2 production due to COX2-specific inhibitor [27] suggested the upregulated expression of COX2 and IL6, as observed previously [5], resulting in testicular inflammation in BPA-exposed testis. In addition to COX2’s roles in regulating inflammation, it is also involved in steroidogenesis and steroid hormone-regulated physiological processes in the testis [6]. Decreased COX2 expression in prenatal BPA-exposed offspring testis (Figure 2) could impact PG synthesis in the testis. Although the PGE2 levels were not measured in the present study, their production might have been altered due to decreased COX2 expression in BPA-exposed adult testis.

We observed that testicular MUFA levels (palmitoleic acid and oleic acid) were significantly increased in BPA-exposed offspring. A decreased level of oleic acid is associated with normal sperm maturation in rats [28], indicating dysregulated sperm maturity in the offspring. The optimal levels of MUFA in the testis also demonstrate successful spermatogenesis since its level correlated negatively with sperm concentration, maturity, regular morphology, vitality, and the fertility index and positively with sperm necrosis [12]. MUFAs serve as unsaturated fatty acid substrate pools that convert to 20- and 22-carbon n-9 PUFAs, further desaturating and elongating in the testis. The elongation capacities for unsaturated fatty acids in the testis are high. The excess accumulation of MUFAs in BPA-exposed testis indicates an inefficient conversion of these unsaturated fatty acids into PUFAs, which might have impacted testis maturity due to an inadequate supply of LCPUFAs.

Similar to BPA-exposed testis in this study, a higher level of palmitoleic acid was associated with increased fat synthesis in BPA-exposed rat adipose [8], and the plasma level of palmitoleic acid correlated with abdominal adiposity in humans [29], indicating BPA’s role in modulating fat metabolism in offspring testis. Again, high SCD (stearoyl-CoA desaturase or delta-9-desaturase) activity is associated with increased fatty acid synthesis, decreased fatty acid oxidation [30], and carries a higher risk of developing metabolic diseases [31]. Since SCD-1 and SCD-2 activities were not induced in response to lower PUFA content of the mature testis [21], this implicates that the induction of these lipogenic enzymes probably occurred due to endocrine or other disorders in the offspring.

Estrogen mimics of BPA might affect spermatogenesis in several ways, including a disruption of the blood–testis barrier [32], widening the interstitial gap between seminiferous tubules [5], impairing testicular steroidogenic hormone activity [33], and others. These mechanisms may coexist in vivo and also impact testicular lipid metabolism, including the changing fatty acid composition in the testis due to increased lipogenesis, as observed in our study. Our data suggest that the balance between testicular fatty acid metabolism (utilization and synthesis) was disrupted in BPA-exposed rats. BPA, an estrogen-mimicking steroid, can easily diffuse into the nucleus across the cell membrane and modulate the transcription of genes [34]. Therefore, BPA exposure can affect the lipogenic regulation of adult offspring testis due to fetal epigenetic aberration of an endocrine–metabolic switch of the male gonads.

The testicular metabolism observed in BPA-exposed rats exhibits similarities to its obesogenic effects demonstrated by disrupting metabolic activity and making the body prone to overweight and obesity. Excess BMI regulates fatty acid metabolism in the testis [18]. Despite the changes in body weight and adiposity [5], lipid profiles such as triglyceride and total cholesterol levels did not change in BPA- and BPS-exposed offspring in plasma and testis (Table 3). Thus, in utero, BPA exposure meditates subtle tissue-specific metabolic changes in regulating the endocrine–metabolic axis of the male reproductive system.

The composition of n-6 LCPUFAs (ARA and DPA) in the testis was decreased in BPA-exposed rats (Table 2). In contrast, increased DPA is associated with normal sperm maturation in rats [28] and may be responsible for dysregulated sperm maturity in the offspring exposed to BPA but not with BPS. Moreover, most LCPUFAs generally decreased except DPA and DTA (22:4 n-6), which remained constant to support the synthesis of very long-chain PUFA during rodent sperm maturity. The rat testicular cells contain high amounts of DPA [35], whereas human testicular cells contain high amounts of DHA. Thus, DPA may perform an equivalent role in rodent testis as DHA does in humans [9]. A decreased DPA level indicates its effects on membrane fluidity and structural development of spermatids in BPA-exposed testis. Decreased DHA and PUFA in spermatozoa are correlated with male infertility [13], indicating potential risks of offspring for infertility due to in utero BPA exposure. Decreased DPA could impair sperm quality since reduced LCPUFAs containing spermatids showed attenuated sperm numbers associated with infertility index in male mice [17]. A DPA-rich lipid moves to surround the spermatid heads when these are embedded in the Sertoli cytoplasm of a healthy mature rat testis [12].

Several enzymes are involved in LCPUFA biosynthesis by desaturation and elongation activities, such as FADS1, FADS2, ELOVL2, ELOVL5, and others. In addition, SCD1, SCD2, FASN, and Acyl Co-A synthetase 6 are also involved in maintaining fatty acid homeostasis in the testis. Despite higher levels of LA, the precursor of n-6 LCPUFAs, the ARA and DPA levels decreased, indicating that the desaturation and elongation of fatty acids were inhibited by BPA exposure. The increased LA and decreased ARA in BPA-exposed testis indicate that both FADS1 and FADS2 are likely to be affected. Again, due to reduced ARA and DTA, DPA was synthesized much less in BPA-exposed rat testis. Although FADS2 is considered the rate-limiting step as it acts twice in the LCPUFA synthesis pathway, FADS2 mRNA expression did not change in BPA-exposed testis (Figure 5) in our study. FADS2 is predominantly expressed in Leydig cells, while FADS1 is expressed in Leydig, Sertoli, and Germ cells of the testis [36]. A higher signal of FADS1 (Figure 7) in the regions around seminiferous tubules where Sertoli and Germ cells are predominant indicates FADS1 could be the predominant desaturase type in the testis of BPA-exposed rats.

In addition to the liver, brain, and adipose, the mammalian testes are also involved in PUFA metabolism. The testis and epididymis also perform specific spermatogenesis roles by modulating LCPUFA metabolism. Viable sperms have a high level of LCPUFAs in their membrane, which maintains sperm fluidity and moves around with a greater velocity. Adequate LCPUFAs are ensured by liver metabolism involving desaturase and elongase enzymes. Table 2 suggests that the whole testis is involved in differential LCPUFA synthesis between control and BPA-exposed rats. While LA, the precursor n-6 PUFAs, intermediator’s metabolites, such as MUFAs (OA), were formed in excess, n-6 LCPUFAs contents were reduced in the testes of the BPA-exposed group.

FADS1 metabolites (such as ARA and their metabolites) are active in Sertoli cells, helping elongate spermatids. FADS1 is involved in the initial step of LCPUFA synthesis, i.e., ARA, by introducing double bonds to LA. Decreased expression of FADS1 in the ARA metabolic pathway of BPA-exposed testis may result in abnormal sperm maturation as the latter depends on this metabolite. The BPA exposure impairs the processes associated with spermatogenesis by modulating fatty acid metabolism in the testis, and the changes were closely similar to HFD-mediated effects [37].

The immunolocalization of FADS1 underlines their active metabolism in the seminiferous tubule. FADS1 is regulated by the transcription factors controlling fatty acid metabolism and lipogenic gene regulation by producing PUFAs. FADS1 is typically detected in elongated spermatids and epididymal cells in animals [36]. A decreased FADS1 expression in BPA-exposed testis failed to add adequate n-6 LCPUFA, such as ARA, during the fusion of the sperm membrane, which might affect sperm competency of the testis. Since ARA is secreted by Sertoli cells in rat testis by an FSH-dependent pathway [38], the estrogen mimic BPA might have dysregulated ARA secretion in their testicular cells in an FSH-dependent manner.

PPARα and PPARβ are expressed in interstitial, germ, and Sertoli cells, while PPARγ expression is specific to Sertoli cells [39]. Testicular Sertoli cells play a vital role in spermatogenesis by supporting germ-cell differentiation, meiosis, and the elongation of spermatozoa. A decreased expression of PPARα and PPARγ in BPA-exposed testis (Figure 3) indicates that the tissue collected in our analysis contains both cell types. The loss of PPARγ in mice oocytes results in impaired fertility [40], while it is adequately expressed in human spermatozoa to maintain energy homeostasis in the testis [41]. BPA exposure could affect sperm energy metabolism due to reduced expression of PPARs. The activation of PPARγ signalling promotes the transcriptional activation of lipid metabolic genes and regulates metabolic energy demands in Sertoli cells. Therefore, the decreased expression of PPARγ in BPA-exposed testis could compromise energy utilization, affect germ cell differentiation, meiosis, spermatids elongation, and dysregulate normal spermatogenesis. In addition, decreased PPARγ expression could affect fertilization in BPA-exposed testis since increased PPARγ expression in ejaculated spermatozoa improves sperm physiology, capacitation, metabolism, and viability [41].

The involvement of testicular cells with the available LCPUFAs depends on the expression of the PPARs [42]. The skewed LCPUFA composition and altered expression of lipid metabolic genes in our study could be mediated by the BPA-activated PPARγ receptor in the testis [43]. BPA could impact testicular fatty acid metabolism by dysregulating the expression of PPARs. Impaired spermatogenesis is associated with the reduced expression of lipid metabolic genes regulated by PPARγ [23]. Prenatal BPA exposure downregulated fatty acid metabolic genes expression, such as FADS1 and SCD1, in this study, suggesting PPARγ mediated regulation of lipid metabolic genes in the offspring testis.

PPAR signalling affects sperm energy capacitation and motility and thus attracts targets for male infertility. The mechanism involving BPA-induced changes in testicular fatty acid composition and PPAR signalling with sperm energy metabolism is not understood yet. The reduced expression of PPARγ and PPARα could be due to the lower availability of ARA and DPA in the testis. The reduced expression of FADS1 in BPA-exposed rat testis might be mediated by the decreased expression of PPARα in producing lower levels of LCPUFAs observed in our study.

Endothelial lipase or hormone-sensitive lipase (LIPE) and lipoprotein lipase (LPL) are expressed in normal testis but perform distinct biological roles. Increased LIPE expression could be responsible for increased testosterone production [44] in BPA-exposed testis, as we observed previously [5]. The LIPE-knockout mice showed an increased expression of FADS1 and SCD-1 and a decreased expression of ELOVL2 in the testis [45]. However, in the present study, BPA-exposed testis exhibited opposite changes in the expression of FADS1, SCD-1, and ELOVL2, along with a decreased synthesis of n-6 LCPUFAs indicating compensatory involvement of LIPE in the mobilization of selected fatty acids to the testis [46]. ELOVL2 controls the level of n-6 LCPUFA (>25 carbon) in the testis, a prerequisite for male fertility and sperm maturation in mice testis [47]. Increased ELOVL2 expression in BPA-exposed testis (Figure 6B) could compensate for reduced FADS1 expression and protect the complete arrest of spermatogenesis. ACSBG2 (acyl Co-A synthetase bubblegum family member 2) catalyzes the activation of long-chain fatty acids (LCFAs) to their acyl-CoA form for membrane synthesis of cellular lipids and beta-oxidation. As sperm formation requires LCFAs and VLCFAs for membrane biogenesis [48], a decreased trend in ACSBG2 expression in our study (Figure 2) may impair fatty acyl CoA activation in spermatogenesis [49].

Fatty-acid-binding protein 9 (FABP9) is required in sperm acrosomal vesicle assembly and its formation. Its absence is associated with increased sperm head abnormalities [50]. In rodents, FABP9 (PERF15) is expressed during spermatogenesis in testicular cells. During spermatid development, FABP9 protein progressively accumulates on the acrosome membrane, facilitating rodent spermiogenesis [51]. The downregulated expression of FABP9 transcript in the BPA-exposed testis (Figure 2) could affect spermatogenesis as FABP9 expression is critically required for germ cell maturation from spermatocytes to spermatids [24]. Moreover, decreased FABP9 expression could impair the structural integrity and stability of the acrosomal membrane in the BPA-exposed offspring testis [52].

FABP4 regulates lipid trafficking and is involved in lipogenesis. BPA-activated PPARγ disturbs lipid metabolism by modulating the expression of FABP4 [43]. Mutant mice (MM-1a) produced round-shaped spermatids but lacked elongated spermatids and exhibited upregulated FABP4 expression in the testis [53]. Thus, BPA-induced FABP4 expression in the testis could affect spermatid formation by modulating lipid metabolism.

Testicular cells produce neutral lipids, store them in lipid droplets (LD) and maintain lipid homeostasis by regulating the expression of LD proteins. LDs control the regulation of lipid storage and oxidation in Sertoli cells. ADRP (PLIN2) is expressed during testis development in rodents and helps maintain scrotal temperature for effective spermatogenesis. Increased ADRP (Figure 1A) could promote heat stress and temperature imbalance in BPA-exposed testis, suppressing spermatogenesis since ADRP expression is upregulated after scrotum heat treatment in mice [54]. Sperm cation channel-like proteins (CATSPER) are exclusively expressed in the testis, where its expression remained maximal at the adult testis and thus plays important roles in sperm functions. Decreased CATSPER2 mRNA expression in BPA-exposed offspring testis (Figure 4) could affect sperm functions since the level of CATSPER2 mRNA expression was significantly downregulated in varicocele-induced rats [55]. Moreover, CATSPER2 transcript level was upregulated in the high-motile spermatozoa than in the low-motile fraction in humans [56], indicating that in utero BPA exposure might have reduced sperm motility by lowering the expression of CATSPER2 in adult offspring testis.

The study is limited to the fact that the PUFA composition in the testis was based on total fatty acids rather than the phospholipid composition. Again, VLC-PUFAs (C ≥ 22) could not be measured in testis due to the unavailability of the standards. Despite changes in the SCD or delta-9-desaturase index, its protein expression could not be measured. While the earlier report suggests that BPA delivery in pregnancy altered fatty acid composition at a concentration of 0.5 µg/kg bw/day [8]; however, BPA (0.4 µg/kg bw/day) with a closely similar concentration in our study did not affect the fatty acid composition of the testis and plasma. The discrepancies could be due to the higher potency of BPA when dissolved in ethanol than olive oil and ad libitum intake of drinking water than controlled gavage delivery used in the present study. Despite these limitations, these data highlight that a disruption in the tightly regulated testicular metabolism due to in utero BPA exposure affected lipogenesis, LCPUFA composition, and desaturase expression in the adult offspring testis, which may affect spermatogenesis associated with bioenergetics, sperm motility, and its function. Earlier, we reported that fetal BP exposure-induced metabolic changes resulted in ROS overproduction and oxidative stress in the testis [5]. Excess ROS produced in the testis can damage sperm DNA and sperm lipids and contribute to decreased sperm quality. The desaturase expression is unaffected in rat testis even during essential fatty acid deficiency [21], but in utero, BPA exposure results in changed FADS1 expression in offspring testis, indicating its greater disruption than the insults from dietary deficiency.

In summary, the LA level is increased relatively higher in BPA-exposed testis than in the control rats. Still, their utilization was less in these testes, indicating reduced metabolism and increased lipogenesis as measured by increased SCD-1 and SCD-2, FABP4, and ADRP expression in the testis. Despite higher MUFAs in BPA-exposed testis, the endogenous conversion to ARA was reduced due to decreased desaturase activities in these testes. Decreased FADS1 and PPARγ expression and reduced ARA production in BPA-exposed testis suggest that FADS1 expression is regulated by the dysregulated expression of transcription factors, PPARγ, and the desaturation of fatty acids. Compared to the control testis, BPA-exposed rats exhibited a dispersed FADS1 signal around ST cells, reduced PPARγ expression and decreased n-6 LCPUFA and FADS1 expression levels. The desaturation/elongation rate of PUFAs is differed by species, sex, endocrine status, and diet but is not reported due to BPA exposure in utero. This first report suggests that, in utero, BPA exposure can affect FADS1 expression in the offspring of the mature testis. The Sertoli cells were possibly affected more than germ cells of the testis because Sertoli cells are more active in LCPUFA metabolism, richer in polyunsaturated fatty acids (PUFAs), and correlate a higher expression of Δ5- (FADS1) and Δ6-desaturase (FADS2). The reduced ARA could have produced fewer metabolites supporting steroidogenesis in BPA-exposed testis.

## 4. Materials and Methods

### 4.1. Animal Experiments

Three-month-old females (n = 40) and males (n = 20) Wistar rats were obtained from the animal facility of the National Institute of Nutrition, India. The rats were housed in pairs per cage at a constantly regulated temperature (22 ± 2 °C) under light-controlled housing conditions (~12:12 h light/dark cycles). The rats were allowed to acclimate for a week and provided with a chow diet and water ad libitum throughout the study. The fertile estrous cycle was determined by using methylene blue staining of vaginal smears. The rats were kept for mating overnight (M:F = 2:1) during their fertile period. The presence of vaginal plugs confirmed mating. After mating, the males were removed, and the day was considered a gestational day (gD) 0. The pregnant female rats were randomly divided into seven groups. Bisphenols (BPA #CAS 80-05-7, BPS #CAS 80-09-1, Sigma Aldrich, St. Louis, MI, USA) were dissolved in extra virgin olive oil to obtain the desired concentrations of 0.4, 4.0, and 40.0 µg/kg body weight per day. The control group was only administered extra virgin olive oil. The pregnant rats were administered bisphenols via oral gavage from gD 4 to 21, as described previously [5]. The Institutional Animal Ethical Committee of ICMR-National Institute of Nutrition approved all of the procedures, which conformed to the guidelines of the Care and Use of Laboratory Animals of the National Institute of Nutrition. The animals were humanely treated and were alleviated of suffering.

### 4.2. Collection of Blood and Testis

The blood was collected in EDTA-coated tubes from the adult rats (90 d) by puncturing the retro-orbital plexus using a sterile capillary tube. The plasma was separated and stored at −80° C for further analysis. The male adult rats were euthanized by carbon dioxide asphyxiation. The testis was removed and washed in sterile PBS to remove traces of blood. Testis was stored in Bouin’s solution and snap-frozen in liquid nitrogen.

### 4.3. Total Fatty Acid Composition

The plasma and testis from adult (90 days) rats were used to estimate the total fatty acid composition. As performed before, the lipids were isolated from the plasma and testis tissue (~100 mg) [57]. The total lipid fraction was methylated by incubating with sulfuric acid (2%) containing methanol and butylated hydroxytoluene (10 mg/L). After cooling the fraction with petroleum ether, fatty acid methyl esters were drawn out. Gas chromatography (Perkin Elmer Clarus 680, Waltham, MA, USA) connected with a silica capillary column (30 m × 0.25 mm × 0.2 μm, Supelco, Bellefonte, PA, USA) analyzed the plasma and testis fatty acid composition. The fatty acid standards based on recognized retention times were employed to identify the unknown samples (Nu-chek-Prep, Elysian, MN, USA).

### 4.4. Triglyceride and Cholesterol Assays

Using a blade homogenizer, the testis tissue (50–100 mg) was homogenized in 5% aqueous Nonidet P40 (NP-40). The homogenate was heated at 80 °C in a water bath to solubilize the lipids. Testicular triglycerides and cholesterol were assayed using an enzymatic kit (Biosystems, Barcelona, Spain).

### 4.5. Real-Time-Polymerase Chain Reaction (RT-qPCR)

Total RNA isolation was carried out using TRIzol reagent (Cat# T9424, Sigma). Briefly, the testis tissue (50 mg) was homogenized in TRIzol solution using zirconia beads (Cat#11079124zx, Biospec, Bartlesville, OK, USA) in a mini bead beater (Biospec). The addition of chloroform achieved phase separation. The upper aqueous phase was separated, and RNA was precipitated using chilled isopropanol. The RNA pellet was dissolved in sterile nuclease-free water. The total RNA was purified from genomic DNA by employing a DNase I kit (Cat#AMPD1, Sigma). Subsequently, Nanodrop (Thermo scientific, ND1000, Waltham, MA, USA) was employed to assess the quality and quantity of the RNA. Next, the total RNA (1 µg) was converted to cDNA using a kit (Cat #1708891 Biorad, Hercules, CA, USA). Finally, qRT-PCR was performed using the Stratagene Mx3005p qPCR system (Agilent Technologies, Santa Clara, CA, USA) using TB green pre-mix Ex-Taq II (Cat#RR208, Takara) and KiCqStart SYBR green primers (Sigma Merck, St. Louis, MI, USA, Appendix A). The obtained Ct values were used for relative quantification of mRNA expression using the ddCt method. The expression of β-actin mRNA was used as endogenous control.

### 4.6. Immunoblotting

The testis lysates were prepared by homogenizing tissue using liquid nitrogen in RIPA buffer containing protease inhibitor cocktail. The supernatant was collected, and the protein was quantified using the BCA method (Pierce™ BCA protein assay kit). An equal quantity of protein was loaded in each well and resolved onto 12% SDS-PAGE gel. After resolving, the proteins were transferred onto a nitrocellulose membrane (#1620112, Bio-Rad) using a wet transfer system. The membrane was blocked with 5% skim milk powder in TBS to avoid non-specific antibody binding. The blots were incubated with primary antibody overnight using corresponding dilutions (Appendix A). The blots were washed thrice with TBST and incubated with HRP conjugated secondary antibody for 90 min at room temperature. Immunoblot signals were captured in the gel imaging system after adding the ECL substrate. The band intensity was quantified as a measure of protein expression using Image J software 1.50i (NIH, Bethesda, MD, USA).

### 4.7. Immunofluorescence

Testis tissues were collected from adult rats (90 d) and fixed in Bouins solution overnight. The fixed tissues were embedded in paraffin, and 4 µm sections were obtained with the help of a microtome. The paraffin sections were mounted onto pre-coated slides. The slides were heated at 60 °C for deparaffinization, followed by a couple of xylene washes. The sections were rehydrated using varying ethanol concentrations and washed with distilled water. Antigen retrieval was achieved by boiling the slides in a microwave oven for 5 min using citrate buffer (pH 6.0). The slides were washed with PBS and blocked with 3% horse serum in a moist, humidified chamber. Primary antibody was added and incubated at 4 °C overnight. The slides were washed with PBS twice and incubated with Alexa fluor conjugated secondary antibody (#A11034, ThermoFischer, Waltham, MA, USA) for 1 h, then mounted with fluoroshield in DAPI medium (#F6057, Sigma, St. Louis, MI, USA). Images were captured using a confocal microscope (DMi8) with SP8 infrared lasers (Leica microsystems).

### 4.8. Statistical Analysis

The data, which are normally distributed, were analyzed by one-way ANOVA followed by post hoc Sidak’s or Tukey’s multiple comparison tests using Graph pad Prism v.8. The data were also calculated by paired *t*-test to compare the two groups. The results were calculated from the data derived from repeated independent experiments mentioned in the text or figure legend. Statistical significance was considered when *p* < 0.05. All experimental data are expressed as mean ± standard error of the mean (SEM).

## 5. Conclusions

In utero BPA exposure could disrupt the fetus’s gonadal programming via dysregulated fatty acid metabolism in the offspring testis. This supports the notion that disrupted testicular development in fetuses could arise due to adverse environmental exposure. Prenatal BPA exposure decreased n-6 LCPUFAs in the testis by modulating the endogenous metabolism of these fatty acids aided with concomitant reduction in the expression of FADS1, which may affect sperm maturity and quality. Thus, fetal BPA exposure reduces testicular long-chain fatty acid availability in adult testis, which may ultimately affect spermatogenesis and pose a risk for male infertility. The development disturbance of the gonads due to prenatal BPA exposure could be linked with fertility impairment due to aberrant changes in the LCPUFA composition of the offspring testis. Thus, lifestyle choices and ubiquitous chemical exposure may affect the reproductive health of adults due to adverse effects on fetal development.

Male fertility is declining globally. In addition to depleted LCPUFAs in the diet, daily exposure to endocrine-disrupting substances could negatively affect spermatogenesis at metabolic and gonadal levels. Testicular dysgenesis syndrome causally originates from adverse environmental exposure, which may disrupt fetal gonadal development and programming due to BPA exposure. Furthermore, the interplay of bisphenol exposure and the endogenous lipid metabolism of the testis could dysregulate the functions of male reproductive systems. However, further studies are required to examine the endocrine-metabolic cross-talk to the early life exposure of this most abundant plasticizer on the male reproductive function in humans.

## Figures and Tables

**Figure 1 ijms-24-03769-f001:**
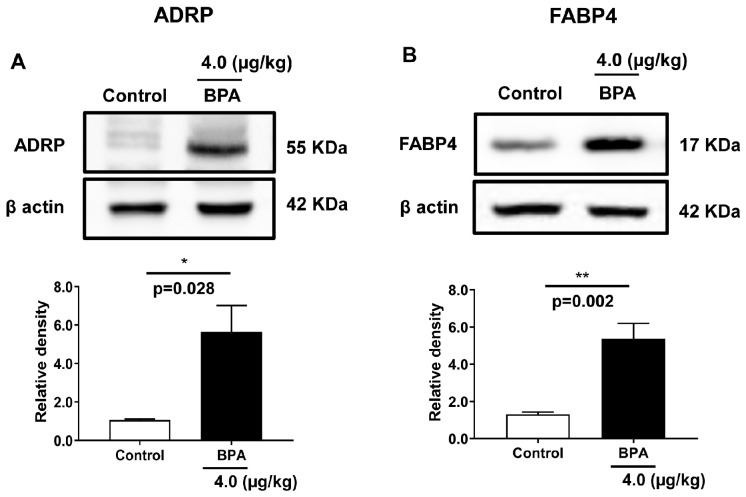
Expression of fatty acid storage and trafficking proteins in the adult offspring testis exposed to BPA in utero. Immunoblots of (**A**) adipose differentiation-related protein (ADRP) and (**B**) fatty-acid-binding protein 4 (FABP4) are derived from the testis homogenate of 90 d rat. Blots show the expression of these proteins. Respective quantitative bars indicate changes in protein levels expressed as the relative band density after normalized with β actin protein expression. Data represent mean ± SEM of three independent experiments. Individual *p*-values are indexed in figures. * *p* < 0.05, ** *p* < 0.005 vs. control (Student’s *t*-test).

**Figure 2 ijms-24-03769-f002:**
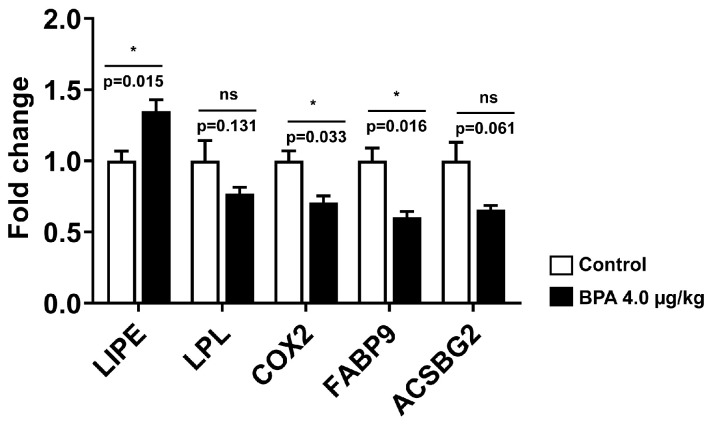
The mRNA expression of genes regulates lipogenesis, growth and signalling molecules of steroidogenesis in adult rat testis whose mothers were exposed to BPA during pregnancy. The expression of hormone-sensitive lipase (LIPE), lipoprotein lipase (LPL), fatty-acid-binding protein 9 (FABP9), cyclooxygenase-2 (COX2), and acyl-CoA synthetase bubblegum family member 2 (ACSBG2) levels were measured in the offspring (90 d) rat testis using RT-qPCR as described in the method. Values are represented as means ± SEM (n = 5–6 rats/group). Individual *p*-values are indexed in figures. * *p* < 0.05 vs. control (Student’s *t*-test). ns: not significant.

**Figure 3 ijms-24-03769-f003:**
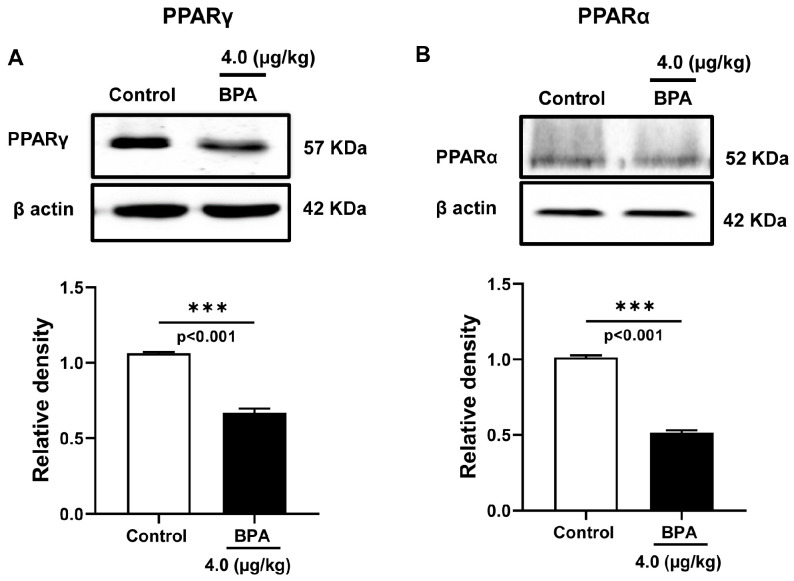
Expression of fatty acid-regulated transcription factors in the adult rat testis exposed to BPA in utero. Immunoblots of (**A**) peroxisome proliferator-activated receptor gamma (PPARγ) and (**B**) peroxisome proliferator-activated receptor alpha (PPARα) were performed with the testis homogenate of 90 d rats. Blots show the expression of these proteins and respective quantitative changes in protein levels expressed as the relative band density after normalized with β actin protein expression. Data represent mean ± SEM of three independent experiments. Individual *p*-values are indexed in figures. *** *p* < 0.0005 vs. control (Student’s *t*-test).

**Figure 4 ijms-24-03769-f004:**
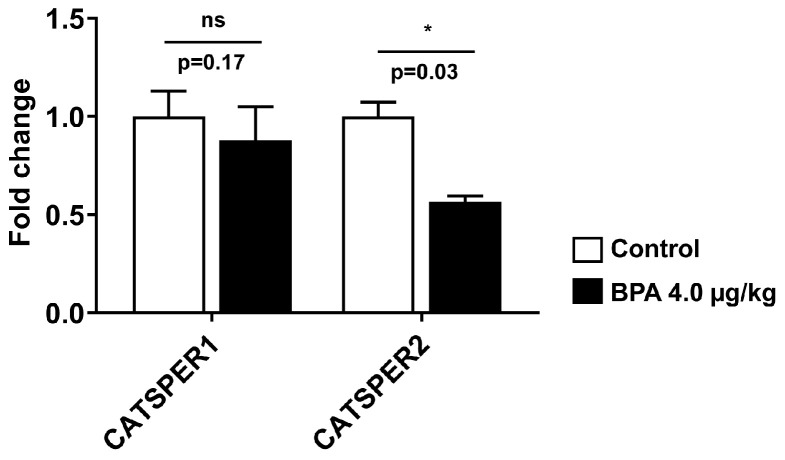
The mRNA expression of genes involved in sperm motility. CATSPER1 and CATSPER2 were measured in rat testis tissue using RT-qPCR after normalizing their expression with the expression of endogenous control β-actin. Values are represented as means ± SEM. Values are expressed as means ± SEM (n = 5–6 rats/group). Individual *p*-values are indexed in figures. * *p* < 0.05 vs. control (Student’s *t*-test). ns: not significant.

**Figure 5 ijms-24-03769-f005:**
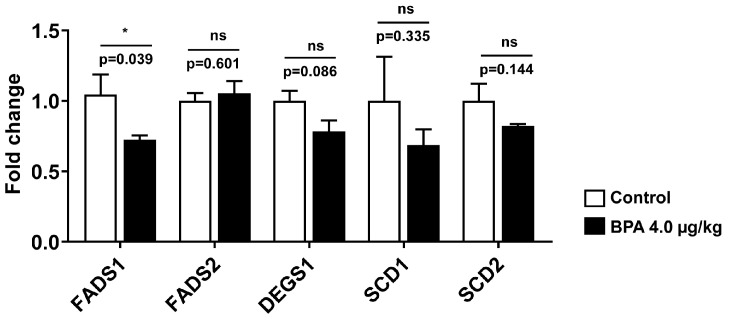
The mRNA expression of genes involved in long-chain-PUFA synthesis (desaturation and elongation) was measured in the testis of 3-month-old rats exposed to BPA in utero. The expression of fatty acid desaturase 1 (FADS1 or delta-5 desaturase), fatty acid desaturase 2 (FADS2 or delta-6 desaturase), delta 4 desaturase sphingolipid 1 (DEGS1), elongation of very long-chain fatty acids-like 2 (ELOVL2), elongation of very long-chain fatty acids-like 5 (ELOVL5), stearoyl-coenzyme A desaturase 2 (SCD2), and stearoyl-coenzyme A desaturase 1 (SCD1) levels were measured using RT-qPCR as described in the method. Values are represented as means ± SEM (n = 5–6 rats/group). Individual *p*-values are indexed in figures. * *p* < 0.05 vs. control (Student’s *t*-test). ns: not significant.

**Figure 6 ijms-24-03769-f006:**
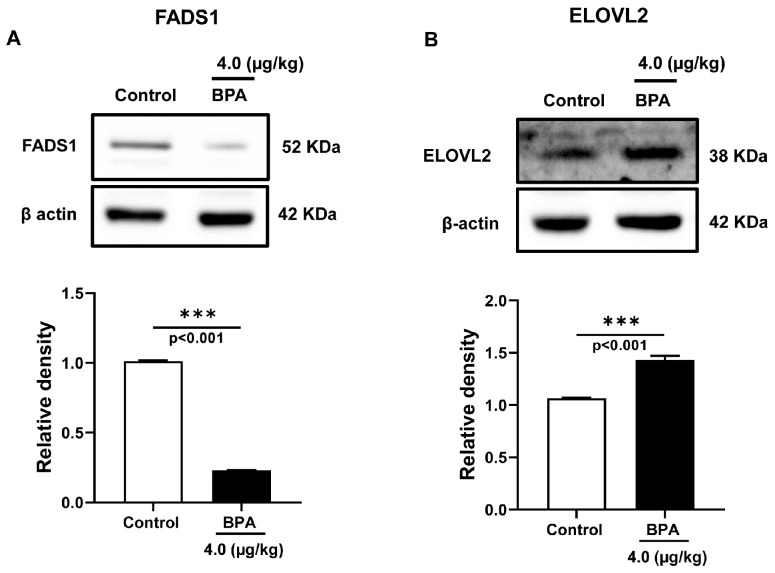
Expression of fatty acid desaturase and elongase in the offspring (90 d) testis exposed to BPA in utero. Immunoblots of (**A**) fatty acid desaturase 1 (FADS1) and (**B**) elongation of very long-chain fatty acids-like 2 (ELOVL2) were performed in testis homogenate of 90 d rats. Blots show the expression of these proteins and respective quantitatively expressed normalized proteins. Data represent mean ± SEM of three independent experiments. Individual *p*-values are indexed in figures. *** *p* < 0.0005 vs. control (Student’s *t*-test).

**Figure 7 ijms-24-03769-f007:**
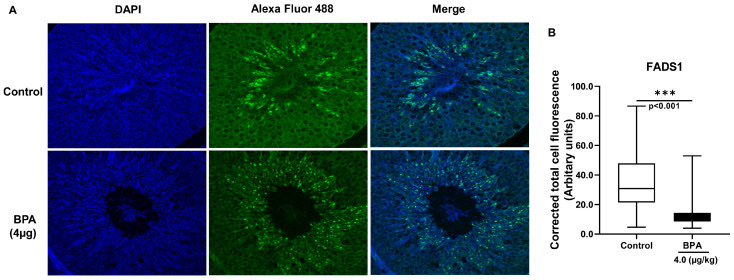
Localization and expression of FADS1 protein by immunofluorescence in the testis of adult Wistar rats that were gestationally exposed to BPA (4.0 µg/kg bw/day). Testis was fixed in Bouin’s solution, dehydrated, and paraffinized. Tissue sections were rehydrated and stained with rabbit anti-FADS1 primary antibody followed by Alexa fluor 488 goat anti-rabbit secondary antibody. (**A**) Immunofluorescence staining of testis tissue sections was performed with Alexa Fluor 488 tagged-FADS1 antibody (green) and nuclei with DAPI (blue). Images were captured with a confocal laser scanning microscope (Leica microsystems, Wetzlar, Germany) at 63 X magnification using an oil immersion objective. (**B**) The total fluorescence of the images was quantified using ImageJ 1.50i (NIH, Bethesda, MD, USA). Corrected total cell fluorescence (CTCF) was calculated after measuring the fluorescence [CTCF = integrated density—(area of selected cell × mean fluorescence of background readings)] and expressed in arbitrary units. Data are expressed as means ± SEM (n = 3). Statistical significance was considered when the *p*-value < 0.05 vs. control (Student’s *t*-test). *** *p* < 0.0005 vs. control (Student’s *t*-test).

**Table 1 ijms-24-03769-t001:** Plasma total fatty acid composition of the adult offspring (90 d) born to dam exposed to bisphenols during gestation (gD4 to gD21) #.

	Experimental Groups (µg/kg bw/day)						
Fatty Acids (nmol%)	Control (0.0)	BPA (0.4)	BPS (0.4)	BPA (4.0)	BPS (4.0)	BPA (40.0)	BPS (40.0)
C16:0 (Palmitic acid)	24.14 ± 0.08 ^a^	23.29 ± 0.29 ^a^	23.66 ± 0.66 ^a^	25.31 ± 1.03 ^a^	24.09 ± 0.22 ^a^	25.95 ± 0.35 ^a^	25.12 ± 0.11 ^a^
C16:1 (Palmitoleic acid)	2.02 ± 0.47 ^a^	1.69 ± 0.41 ^a^	1.32 ± 0.11 ^a^	1.59 ± 0.41 ^a^	1.23 ± 0.03 ^a^	1.72 ± 0.31 ^a^	1.33 ± 0.04 ^a^
C18:1 (Stearic acid)	11.65 ± 0.48 ^a^	11.27 ± 0.46 ^a^	10.65 ± 0.41 ^a^	14.59 ± 1.95 ^a^	11.47 ± 0.45 ^a^	11.42 ± 0.18 ^a^	11.81 ± 0.03 ^a^
C18:1 (Oleic acid)	15.46 ± 0.04 ^a^	16.17 ± 1.22 ^a^	16.22 ± 0.21 ^a^	13.81 ± 0.33 ^a^	13.86 ± 0.58 ^a^	14.50 ± 0.14 ^a^	14.13 ± 0.61 ^a^
C18:2 n-6 (Linoleic acid)	23.99 ± 0.98 ^a^	25.21 ± 0.57 ^a^	26.32 ± 0.80 ^a^	22.13 ± 1.42 ^a^	23.98 ± 0.19 ^a^	22.76 ± 0.09 ^a^	22.94 ± 0.22 ^a^
C20:4 n-6 (Arachidonic acid)	17.82 ± 0.61 ^a^	18.63 ± 2.51 ^a^	18.13 ± 0.51 ^a^	19.37 ± 1.73 ^a^	22.43 ± 0.28 ^a^	19.86 ± 0.46 ^a^	21.32 ± 0.81 ^a^
C22:4 n-6 (Docosatetraenoic acid)	0.10 ± 0.06 ^a^	0.15 ± 0.08 ^a^	0.26 ± 0.05 ^a^	0.07 ± 0.02 ^a^	0.20 ± 0.05 ^a^	0.18 ± 0.01 ^a^	0.22 ± 0.03 ^a^
C22:5 n-6 (Docosapentaenoic acid)	0.26 ± 0.06 ^a^	0.38 ± 0.06 ^a^	0.39 ± 0.07 ^a^	0.19 ± 0.03 ^a^	0.29 ± 0.04 ^a^	0.40 ± 0.03 ^a^	0.37 ± 0.03 ^a^
C22:6 n-3 (Docosahexaenoic acid)	0.55 ± 0.12 ^a^	0.82 ± 0.07 ^ab^	0.97 ± 0.10 ^b^	0.69 ± 0.13 ^a^	0.93 ± 0.03 ^a^	0.82 ± 0.01 ^a^	0.83 ± 0.07 ^a^

# Data were analysed with one-way ANOVA with post hoc Sidak’s multiple comparison test. Values are represented as means ± SEM (n = 3). The experimental group with unlike letters is significantly different at *p* < 0.05 vs. control.

**Table 2 ijms-24-03769-t002:** Testis total fatty acid composition of adult male offspring (90 d) born to dam those exposed to bisphenols during gestation (gD4 to gD21) #.

	Experimental Groups (µg/kg bw/day)				
Fatty Acids (nmol%)	Control (0.0)	BPA (0.4)	BPS (0.4)	BPA (4.0)	BPS (4.0)
C16:0 (Palmitic acid)	38.34 ± 0.26 ^a^	40.42 ± 0.26 ^a^	38.91 ± 0.43 ^a^	22.31 ± 1.88 ^b^	38.73 ± 0.80 ^a^
C16:1 (Palmitoleic acid)	1.37 ± 0.08 ^a^	1.03 ± 0.25 ^a^	1.15 ± 0.12 ^a^	9.35 ± 0.76 ^b^	1.46 ± 0.32 ^a^
C18:0 (Stearic acid)	8.41 ± 0.23 ^a^	8.11 ± 0.26 ^a^	8.24 ± 0.38 ^a^	3.11 ± 0.12 ^b^	8.11 ± 0.27 ^a^
C18:1 (Oleic acid)	12.20 ± 0.36 ^a^	10.55 ± 0.07 ^a^	11.19 ± 0.42 ^a^	29.39 ± 0.86 ^b^	12.44 ± 0.66 ^a^
C18:2 n-6 (Linoleic acid)	5.03 ± 0.16 ^a^	4.08 ± 0.22 ^a^	4.70 ± 0.52 ^a^	24.75 ± 1.02 ^b^	5.30 ± 0.95 ^a^
C20:4 n-6 (Arachidonic acid)	14.64 ± 0.44 ^a^	13.96 ± 0.10 ^a^	14.47 ± 0.23 ^a^	4.24 ± 0.40 ^b^	14.20 ± 0.71 ^a^
C22:4 n-6 (Docosatetraenoic acid)	1.84 ± 0.01 ^a^	1.84 ± 0.03 ^a^	1.81 ± 0.12 ^a^	0.89 ± 0.08 ^b^	1.81 ± 0.23 ^a^
C22:5 n-6 (Docosapentaenoic acid)	16.14 ± 0.61 ^a^	17.84 ± 0.31 ^a^	17.97 ± 0.36 ^a^	2.56 ± 0.64 ^b^	16.55 ± 0.37 ^a^
C22:6 n-3 (Docosahexaenoic acid)	0.23 ± 0.03 ^a^	0.26 ± 0.02 ^a^	0.31 ± 0.01 ^a^	0.24 ± 0.07 ^a^	0.29 ± 0.01 ^a^
SCD-16 (C16:1/C16:0)	0.03 ± 0.00 ^a^	0.02 ± 0.00 ^a^	0.02 ± 0.00 ^a^	0.43 ± 0.06 ^b^	0.03 ± 0.00 ^a^
SCD-18 (C18:1/C18:0)	1.45 ± 0.06 ^a^	1.30 ± 0.03 ^a^	1.36 ± 0.09 ^a^	9.42 ± 0.57 ^b^	1.54 ± 0.13 ^a^
Desaturation index (C20:4 n-6/C18:2 n-6)	2.92 ± 0.15 ^a^	3.45 ± 0.17 ^a^	3.19 ± 0.34 ^a^	0.17 ± 0.02 ^b^	2.97 ± 0.55 ^a^

# Data were analysed by one-way ANOVA with post hoc Sidak’s multiple comparison test. Values are represented as means ± SEM (n = 4). The experimental groups with unlike letters are significantly different at *p* < 0.05 vs. control and BPA vs. BPS of similar concentration.

**Table 3 ijms-24-03769-t003:** Triglycerides and cholesterol contents of adult rat testis (90 d) exposed to BPA from gD 4 to gD 21.

	Experimental Groups (µg/kg bw/day)		
Lipid Content	Control (0.0)	BPA (4.0)	*p* Value
**Plasma (mmol/L)**			
Triglycerides, TG	1.59 ± 0.03	1.53 ± 0.03	0.27
Total cholesterol, TC	2.90 ± 0.02	2.71 ± 0.08	0.08
**Testis (mg/g tissue)**			
Triglycerides, TG	4.05 ± 0.05	3.04 ± 0.57	0.34
Total cholesterol, TC	1.00 ± 0.10	1.02 ± 0.12	0.91

Data were analysed with unpaired *t*-test. Values are represented as means ± SEM. Plasma (n = 7–8) and testis (n = 3–5). *p* < 0.05 was considered as statistically significant vs. control.

## Data Availability

All of the data generated or analyzed during this study are available from the corresponding author upon request.

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
