# Peer review of "Fetal Exposure to Endocrine Disrupting-Bisphenol A (BPA) Alters Testicular Fatty Acid Metabolism in the Adult Offspring: Relevance to Sperm Maturation and Quality"

_ijms, 2023, doi:10.3390/ijms24043769_

Round 1
Reviewer 1 Report
I have read with interest the present paper regarding the testicular effects of in-utero BPA exposure of wistar rats. The main observed effect was the dysregulation of testicular fatty acid composition and metabolism in the offspring testis. Although the hypothesis of testicular function disruption by environmental contaminating agents dates back from many years, this paper brings interesting new evidence on the testicular effects of BPA highlighting its association with endogenous testicular lipids, and suggests several mechanisms of disturbances of spermatogenesis and male reproductive function.
The paper is well written and relatively easy to read. Methodologically, I do not see particular flaws in the design and results appear robust and convincing.
The discussion also highlights several associated pathophysiological mechanisms and appears well up-to-date.
I only suggest the authors to address a few minor issues:
- I would further stress in the conclusions that due to possible differences between the animal model and in vivo human effects of BPA exposure, results cannot be directly generalized to humans
- lines 110-111 – spermatogenesis starts from the basal membrane germ cells. Did you mean spermiogenesis?
- lines 567-570 – please remove this text
- the apex letters in the tables should be better explained. Significances are not easy to interpret. Also in table 2 the meaning of the bold numbers should be explained (or the bold should be removed)
- Another possible interpretation of the testicular metabolism (and further spermatogenesis impairment) is that BPA affects the Sertoli cell function and their tight junctions, ultimately disrupting the blood testis barrier (this is well described in literature for example DOI: 10.1016/j.reprotox.2022.06.004) or impairing testicular steroidogenic hormone activity (for example DOI: 10.1016/j.chemosphere.2020.127880). these mechanism are likely to coexist in vivo an might also impact on testicular lipid metabolism. Can you please comment on this?
Author Response
I have read with interest the present paper regarding the testicular effects of in-utero BPA exposure of wistar rats. The main observed effect was the dysregulation of testicular fatty acid composition and metabolism in the offspring testis. Although the hypothesis of testicular function disruption by environmental contaminating agents dates back from many years, this paper brings interesting new evidence on the testicular effects of BPA highlighting its association with endogenous testicular lipids, and suggests several mechanisms of disturbances of spermatogenesis and male reproductive function.
The paper is well written and relatively easy to read. Methodologically, I do not see particular flaws in the design and results appear robust and convincing.
The discussion also highlights several associated pathophysiological mechanisms and appears well up-to-date.
I only suggest the authors to address a few minor issues:
- I would further stress in the conclusions that due to possible differences between the animal model and in vivo human effects of BPA exposure, results cannot be directly generalized to humans
Response: We thank you for your appreciation that manuscript reads well. In the human testis, the n-3 fatty acid pathway preserves several functions associated with male fertility. In contrast, the n-6 fatty acid pathway plays an equivalent role in rodents, as evidenced by several studies. Despite the differences in the predominant fatty acid pathways in testis between these two species, the enzymatic machinery involved in converting saturated fatty acid to long-chain PUFA are closely similar. However, we agree that data from the animal model has no parallel with in vivo human effects on BPA exposure, as we mentioned in the last statement of the conclusion.
- lines 110-111 – spermatogenesis starts from the basal membrane germ cells. Did you mean spermiogenesis?
Response: Thank you for pointing this out. We have revised it.
- lines 567-570 – please remove this text
Response: Lines 567-570 have been deleted.
- the apex letters in the tables should be better explained. Significances are not easy to interpret. Also, in table 2 the meaning of the bold numbers should be explained (or the bold should be removed)
Response: We have explained the table footnote with additional information. The experimental groups with unlike letters are significantly different at p<0.05 vs control and BPA vs BPS of similar concentration. The Bold font has been removed. The legend of table 2. is now revised.
- Another possible interpretation of the testicular metabolism (and further spermatogenesis impairment) is that BPA affects the Sertoli cell function and their tight junctions, ultimately disrupting the blood testis barrier (this is well described in literature for example DOI: 10.1016/j.reprotox.2022.06.004) or impairing testicular steroidogenic hormone activity (for example DOI: 10.1016/j.chemosphere.2020.127880). these mechanism are likely to coexist in vivo an might also impact on testicular lipid metabolism. Can you please comment on this?
Response: Thank you for your critical insight. Multiple mechanisms may coexist in vivo and also impact testicular lipid metabolism. We have added these points and citations in the revised manuscript.
Reviewer 2 Report
Building on observations of adult testicular irregularities in physiology and metabolism following in utero exposures to BPA, the authors examined, in considerable detail, the effects of similar gestational exposures to BPA and BPS on several measures of lipid metabolism and steroidogenesis in adult male rats. This is an excellent and comprehensive series of studies examining a number of lipid metabolic pathways that are ultimately related to spermatogenesis and fertility success.
The methods are appropriate; the number of different and relevant assays are impressive; the analyses and presentation of results are clear. And the discussion provides a complex integrated narrative of the importance of the data from this series of studies. That equivalent doses of prenatal exposure to BPS did not mimic the effects of gestational BPA exposures provided an internal negative control for the BPA effects.
All-in-all, this is an excellent paper which, after a little editing should be accepted for publication without additional revision. The editing is just to be careful that long, complex sentences are really clearly written. (Additionally, the paragraph in lines 567-570 should be deleted.)
Author Response
Building on observations of adult testicular irregularities in physiology and metabolism following in utero exposures to BPA, the authors examined, in considerable detail, the effects of similar gestational exposures to BPA and BPS on several measures of lipid metabolism and steroidogenesis in adult male rats. This is an excellent and comprehensive series of studies examining a number of lipid metabolic pathways that are ultimately related to spermatogenesis and fertility success.
The methods are appropriate; the number of different and relevant assays are impressive; the analyses and presentation of results are clear. And the discussion provides a complex integrated narrative of the importance of the data from this series of studies. That equivalent doses of prenatal exposure to BPS did not mimic the effects of gestational BPA exposures provided an internal negative control for the BPA effects.
All-in-all, this is an excellent paper which, after a little editing should be accepted for publication without additional revision. The editing is just to be careful that long, complex sentences are really clearly written. (Additionally, the paragraph in lines 567-570 should be deleted.)
Response: We thank you for your appreciation that manuscript reads well. Few statements state many related events in the discussion, and thus became long.
The paragraph with lines 567-570 has now been deleted.
Reviewer 3 Report
The manuscript entitled "Fetal exposure to endocrine disrupting-bisphenol A (BPA) alters testicular fatty acid metabolism in the adult offspring: relevance to sperm maturation and quality" presents very interesting findings regarding the BPA exposure that could affect endogenous long-chain fatty acid metabolism and steroidogenesis in the adult testis, which might dysregulate sperm maturation and quality. Therefore, iI suggest the article be accepted as submitted.
Author Response
The manuscript entitled "Fetal exposure to endocrine disrupting-bisphenol A (BPA) alters testicular fatty acid metabolism in the adult offspring: relevance to sperm maturation and quality" presents very interesting findings regarding the BPA exposure that could affect endogenous long-chain fatty acid metabolism and steroidogenesis in the adult testis, which might dysregulate sperm maturation and quality. Therefore, iI suggest the article be accepted as submitted.
Response: Thank you for your feedback on accepting our manuscript.